# Impact of Glucose, Inflammation and Phytochemicals on *ACE2*, *TMPRSS2* and Glucose Transporter Gene Expression in Human Intestinal Cells

**DOI:** 10.3390/antiox14030253

**Published:** 2025-02-21

**Authors:** Rizliya Visvanathan, Michael J. Houghton, Gary Williamson

**Affiliations:** 1Department of Nutrition, Dietetics and Food, BASE Facility, Monash University, Level 1, 264 Ferntree Gully Road, Notting Hill, VIC 3168, Australia; 2Victorian Heart Institute, Monash University, Level 2, Victorian Heart Hospital, 631 Blackburn Road, Clayton, VIC 3168, Australia

**Keywords:** type 2 diabetes, COVID-19, genistein, apigenin, artemisinin, sulforaphane

## Abstract

Inflammation is associated with the pathophysiology of type 2 diabetes and COVID-19. Phytochemicals have the potential to modulate inflammation, expression of SARS-CoV-2 viral entry receptors (angiotensin-converting enzyme 2 (ACE2) and transmembrane protease, serine 2 (TMPRSS2)) and glucose transport in the gut. This study assessed the impact of phytochemicals on these processes. We screened 12 phytochemicals alongside 10 pharmaceuticals and three plant extracts, selected for known or hypothesised effects on the SARS-CoV-2 receptors and COVID-19 risk, for their effects on the expression of *ACE2* or *TMPRSS2* in differentiated Caco-2/TC7 human intestinal epithelial cells. Genistein, apigenin, artemisinin and sulforaphane were the most promising ones, as assessed by the downregulation of *TMPRSS2*, and thus they were used in subsequent experiments. The cells were then co-stimulated with pro-inflammatory cytokines interleukin-1 beta (IL-1β) and tumour necrosis factor-alpha (TNF-α) for ≤168 h to induce inflammation, which are known to induce multiple pathways, including the nuclear factor kappa-light-chain-enhancer of activated B cells (NF-κB) pathway. Target gene expression (*ACE2*, *TMPRSS2*, *SGLT1* (sodium-dependent glucose transporter 1) and *GLUT2* (glucose transporter 2)) was measured by droplet digital PCR, while interleukin-1 (IL-6), interleukin-1 (IL-8) and ACE2 proteins were assessed using ELISA in both normal and inflamed cells. IL-1β and TNF-α treatment upregulated *ACE2*, *TMPRSS2* and *SGLT1* gene expression. *ACE2* increased with the duration of cytokine exposure, coupled with a significant decrease in IL-8, *SGLT1* and *TMPRSS2* over time. Pearson correlation analysis revealed that the increase in *ACE2* was strongly associated with a decrease in IL-8 (*r* = −0.77, *p* < 0.01). The regulation of *SGLT1* gene expression followed the same pattern as *TMPRSS2*, implying a common mechanism. Although none of the phytochemicals decreased inflammation-induced IL-8 secretion, genistein normalised inflammation-induced increases in *SGLT1* and *TMPRSS2*. The association between *TMPRSS2* and *SGLT1* gene expression, which is particularly evident in inflammatory conditions, suggests a common regulatory pathway. Genistein downregulated the inflammation-induced increase in *SGLT1* and *TMPRSS2*, which may help lower the postprandial glycaemic response and COVID-19 risk or severity in healthy individuals and those with metabolic disorders.

## 1. Introduction

Intestinal glucose absorption is predominantly mediated by the active and passive glucose transporters sodium-dependent glucose transporter 1 (SGLT1) and glucose transporter 2 (GLUT2) [1]. Overexpression of these genes may increase intestinal glucose absorption, contributing to postprandial hyperglycaemia, a significant risk factor for type 2 diabetes [2,3]. While dietary carbohydrate intake has been shown to modulate glucose transporter expression [4,5,6,7], emerging evidence suggests that inflammation also plays a critical role in regulating intestinal glucose absorption [8,9]. For example, gut inflammation has been shown to upregulate SGLT1 expression in mice [8]. However, the mechanistic interplay between inflammatory signalling and glucose transporter regulation in the intestine remains poorly understood.

Chronic low-grade inflammation is a hallmark feature of metabolic disorders, including obesity and type 2 diabetes [10,11]. Pro-inflammatory cytokines, such as interleukin-1 beta (IL-1β), tumour necrosis factor-alpha (TNF-α) and interleukin-6 (IL-6), are elevated in patients with diabetes [10,11,12] and have been shown to alter glucose metabolism through various signalling pathways [13]. One key mechanism involves dysregulation of the renin-angiotensin system, particularly the angiotensin-converting enzyme 2/angiotensin-(1-7)/Mas receptor (ACE2/Ang-(1-7)/Mas) axis, which is implicated in altered glucose metabolism in diabetes [14,15]. ACE2, which is widely expressed in the gut, regulates intestinal glucose transport by modulating SGLT1 and GLUT2 expression [15,16]. ACE2-derived Ang-(1-7) binding to the Mas receptor inhibits glucose absorption by suppressing SGLT1 and GLUT2 expression while exerting anti-inflammatory effects via tryptophan metabolism [15]. Therefore, inflammation-induced downregulation of ACE2 may exacerbate dysregulated glucose absorption, further contributing to metabolic dysfunction.

Dietary phytochemicals have gained interest as potential therapeutic agents for metabolic disorders due to their anti-inflammatory, antioxidant and glucose-lowering properties [17,18]. Polyphenols such as genistein (an isoflavone from soybeans), apigenin (a flavone found in vegetables and herbs), artemisinin (a sesquiterpene lactone with anti-malarial and anti-inflammatory properties [19]) and sulforaphane (an isothiocyanate from cruciferous vegetables) have demonstrated regulatory effects on glucose transport and inflammation in different tissues [18,19,20,21,22,23], thereby presenting a potential intervention strategy for managing inflammation-induced changes in glucose metabolism. However, their precise effects on glucose transport in inflamed intestinal tissues remain to be fully elucidated.

Although numerous studies on the regulation of glucose transporters by polyphenols have been published, to our knowledge, studies have yet to investigate the role of ACE2 in this mechanism. In this work, we aimed to look at the link between inflammation, ACE2 and the glucose transporters SGLT1 and GLUT2, and the potential for phytochemicals to normalise the changes brought about by inflammation. Additionally, given the dual role of ACE2 in glucose metabolism and as a SARS-CoV-2 viral entry receptor, alongside transmembrane protease, serine 2 (TMPRSS2) [24,25,26], this study also examined the effect of glucose, inflammation and phytochemicals on the expression of these viral entry receptors as a secondary outcome.

## 2. Materials and Methods

### 2.1. Chemicals and Reagents

The Caco-2/TC7 cells were a kind donation from the Rousset Lab (U178 INSERM, Vilejuif, France). Genistein, apigenin and sulforaphane were purchased from Merck Life Science Pty Ltd. (Bayswater, VIC, Australia). A Coomassie Plus Bradford assay kit and bovine serum albumin standards were purchased from Thermo Fisher Scientific, Inc. (Scoresby, VIC, Australia). Human ACE2 (ab235649) and human IL-8 (ab100575) ELISA kits were purchased from Abcam Australia Pty Ltd. (Melbourne, VIC, Australia). A human IL-6 and IL-8 cytokine panel multiplex ELISA kit (HCYTMAG-60K) was procured from EMD Millipore (Darmstadt, Germany). An Aurum Total RNA Mini Kit for RNA extraction from cells, a QX200 droplet digital PCR (ddPCR) system, ddPCR Supermix for Probes (no dUTP) and all other materials used for ddPCR were purchased from Bio-Rad Laboratories (South Granville, NSW, Australia). The high-capacity RNA-to-cDNA reverse transcription kits and FAM™-labelled or VIC™-labelled TaqMan primers for *ACE2* (Hs01085333_m1), *SGLT1* (Hs01573793_m1), *GLUT2* (Hs01096908_m1), *TMPRSS2* (Hs01122322_m1), *ACE* (Hs00174179_m1) and *TBP* (TATA-box binding protein, Hs00427620_m1) were from Thermo Fisher Scientific. Human IL-1β and TNF-α proteins and all cell culture vessels, as well as other chemicals and reagents, were purchased from Merck Life Science unless specified otherwise. High-purity (18.2 MΩ/cm) milliQ water (Merck Life Science) was used throughout. All of the chemicals and reagents used were of the highest purity.

### 2.2. Cell Culture

Human colon carcinoma-derived Caco-2/TC7 cells were employed because they develop into a more homogenous population of cells with traits akin to human enterocytes [27]. The cells were grown in 75 cm^2^ culture flasks in complete Dulbecco’s modified Eagle medium (DMEM) composed of high glucose (4.5 g/L or 25 mM) supplemented with 20% (*v*/*v*) heat-inactivated foetal bovine serum (FBS; Thermo Fisher Scientific), 2% (*v*/*v*) nonessential amino acids (NEAAs), 2% (*v*/*v*) Glutamax (Thermo Fisher Scientific) and 1% (*v*/*v*) penicillin-streptomycin (final concentrations of 100 U/mL penicillin and 100 mg/mL streptomycin). Cells were seeded at a density of 0.2 × 10^5^ cells/cm^2^ and sub-cultured at ~60% confluence. Using a 0.25% (*v*/*v*) trypsin-EDTA solution, the cells were carefully lifted from the flask and then cultured at a density of 0.2 × 10^5^ cells/cm^2^ in new 75 cm^2^ cell culture flasks. Cells were maintained in an incubator at 37 °C in a humidified 10% CO_2_/90% air (*v*/*v*) atmosphere. Cell passages between 32 and 40 were used in the experiments. For a detailed report on Caco-2/TC7 culturing, refer to Barber et al. [28].

### 2.3. Cell Differentiation

For differentiation, cells were seeded on the apical chamber of either 6-well or 12-well Transwell plates with polyester filters (0.4 µm pore size, 24 mm (6-well) or 12 mm (12-well) diameter) at a density of 0.1 × 10^5^ cells/cm^2^ in complete DMEM medium (25 mM glucose) supplemented with 20% (*v*/*v*) FBS in both apical and basolateral compartments. The cells were examined under a microscope daily until they attained 100% confluence, which was noted as day 0. The cells were differentiated until day 7 in the same medium, after which the apical compartment received FBS-free medium, and the basolateral chamber received complete DMEM containing either 10% or 20% (*v*/*v*) FBS. The cells were allowed to differentiate for 14 days with regular medium changes three times a week.

### 2.4. Treatment with Cytokines (Inflammation Model Optimisation Study)

Caco-2/TC7 cells were differentiated in 12-well Transwell plates in 25 mM glucose-containing complete DMEM medium with 20% (*v*/*v*) FBS in the apical and basolateral compartments until day 7. From day 7, the cells were maintained in either 5.5 mM or 25 mM glucose-containing DMEM media, which were FBS-free in the apical compartment and had 10% (*v*/*v*) FBS in the basolateral compartment. Inflammation was induced using a cytokine mixture containing 25 ng/mL IL-1β and 50 ng/mL TNF-α. Differentiated Caco-2/TC7 cells were exposed to the cytokine cocktail for the final 24 h, 48 h, 72 h or 168 h. The cytokines were dissolved in DMEM medium (5.5 mM or 25 mM glucose) containing 10% (*v*/*v*) FBS and introduced into the basolateral compartment daily until day 12. On day 13, the cytokine mixture was prepared in either 5.5 mM or 25 mM glucose-containing FBS-free media to avoid protein interference in the IL-6 and IL-8 estimation. On day 14, apical and basolateral culture media were collected for IL-6 and IL-8 estimation, while the cells were washed in cold PBS three times and RNA was collected in Total RNA Lysis Solution supplemented with 1% (*v*/*v*) β-mercaptoethanol (Bio-Rad) for gene expression studies. The samples were stored at −80 °C until analysis.

### 2.5. Transepithelial Electrical Resistance (TEER) Measurement

Intestinal cell monolayer integrity was assessed by measuring the transepithelial electrical resistance (TEER) using a Millicell-ERS voltohmmeter (Millipore, Bedford, MA, USA). The TEER values were measured before adding the cytokine cocktail (0 h) and following 24 h, 48 h, 72 h and 168 h of cytokine treatment. The electrical resistance was measured at three different locations surrounding the wells. Only monolayers with an epithelial resistance >200 Ω·cm^2^ were used for the experiments [29].

### 2.6. Trypan Blue Exclusion Cell Viability Assay

As previously described [30], the viability of the Caco-2/TC7 cells exposed to the cytokine cocktail was assessed using the Trypan Blue exclusion assay. On day 14, the cytokine-treated cells were washed with warm PBS twice and lifted with 150 µL of 0.25% (*v*/*v*) trypsin-EDTA (12-well Transwell plates) by incubating them for 4–5 min. Next, 150 µL of either 5.5 mM or 25 mM glucose-containing DMEM with 10% (*v*/*v*) FBS was added to the wells, according to their prior treatment, to neutralise the trypsin-EDTA, and an aliquot of the collected cell suspension was mixed with 0.04% (*v*/*v*) Trypan Blue-PBS solution (1:1 (*v*/*v*)). The resuspended cells were counted using a TC20 automated cell counter (Bio-Rad), and data were presented as the percentage of viable cells (Trypan Blue-free cells).

### 2.7. Treatment with Phytochemicals

Caco-2/TC7 cells were seeded and differentiated as described in Section 2.2 and Section 2.3. The cells were differentiated in 25 mM glucose until day 7 and then in 5.5 mM glucose until day 14. The phytochemicals were dissolved in DMSO to prepare a 20 mM stock of sulforaphane and 25 mM of genistein, apigenin and artemisinin stock solutions. Test concentrations of the phytochemicals (sulforaphane = 5, 10 and 20 µM; the rest = 5, 10 and 25 µM) were prepared by dissolving the stock in FBS-free 5.5 mM glucose-containing DMEM media. The final DMSO concentration in all of the test samples and the DMSO vehicle control was 0.1% (*v*/*v*). The apical compartment received the phytochemicals, and the entire treatment duration was either 4 h or 60 h. The cells were treated with the phytochemicals or DMSO (vehicle control) twice daily, from the morning of day 12 for the 60-h treatment or the evening of day 14 for the 4-h treatment. The final addition in the 60-h regimen was 4 h prior to the end of the experiment to match the acute treatment. In the inflamed cell model, Caco-2/TC7 cells were stimulated with the cytokine cocktail containing IL-1β (25 ng/mL) and TNF-α (50 ng/mL) for 72 h (as described in Section 2.4; cytokine exposure duration was selected after considering the target gene (*ACE2*, *SGLT1*, *GLUT2* or *TMPRSS2*) expression pattern from the inflammation model optimisation study). The cytokine mixture was added to the basolateral compartment from the evening of day 11. All cell treatments ended simultaneously. At the end of the experiment, culture media from the basolateral compartments were collected in the inflamed cell model for IL-8 estimation. Cells were washed with chilled PBS thrice, and RNA and protein were collected for gene and protein expression studies. The samples were stored at −80 °C until analysis.

### 2.8. Gene Expression Analysis with Droplet Digital PCR (ddPCR)

Total RNA extractions were performed using the Aurum™ Total RNA Mini Kit (Bio-Rad), following the manufacturer’s instructions. The concentration and quality of the extracted RNA were determined using a Nanodrop 2000ND spectrophotometer (Thermo Fisher Scientific). Reverse transcription was performed using the High-Capacity RNA-to-cDNA reverse transcription kit (Thermo Fisher Scientific), following the manufacturer’s instructions. Quantitative gene expression was performed using a QX200 droplet digital PCR (ddPCR) system (Bio-Rad) in a 96-well plate. Each assay (20 µL) contained 8 μL cDNA (34.1 ng) diluted with RNAse- and DNAse-free water, 1 μL FAM-labelled TaqMan primer (*ACE*, *ACE2*, *SGLT1*, *GLUT2* or *TMPRSS2*), 1 μL VIC-labelled TaqMan primer *TBP* and 10 μL ddPCR supermix for probes (no dUTP) or various controls with each component absent. Assays were run in duplicate. The ddPCR reaction mix was partitioned into 1 nL-sized droplets using a QX200 droplet generator following the manufacturer’s instructions, which were transferred to a 96-well ddPCR plate for PCR on a C1000 Touch™ thermal cycler (Bio-Rad) and read with a QX200 droplet reader. *TBP* was utilised as a reference to assess performance between ddPCR runs. The data were analysed using the QuantaSoft programme (Bio-Rad). The average number of accepted droplets was >12,000 for all samples. Target DNA concentration (copies/μL) was calculated using Poisson distribution analysis.

### 2.9. Measuring Secretion of Inflammatory Markers IL-6 and IL-8

Cells were treated with the cytokine mixture for different durations for the inflammation model optimisation study as mentioned in Section 2.4. For the phytochemical study (Section 2.7), the cells were co-treated with the cytokine mixture (72 h) and the phytochemicals (4 and 60 h). For the final day of treatment, FBS-free medium was used in both cell compartments to avoid protein interference in IL-6 and IL-8 estimation. At the end of the experiment, cell culture media from the basolateral compartments were collected and stored at −80 °C until analysis. For analysis, the samples were first thawed on ice and centrifuged at 500× *g* for 5 min, and the cell debris-free supernatant was transferred into a pre-labelled fresh tube. For the inflammation model optimisation study, the concentrations of IL-6 and IL-8 (HCYTMAG-60K, EMD Millipore, Darmstadt, Germany) in the culture media were determined using a magnetic bead-based multiplex ELISA system (Magpix, Luminex, Austin, TX, USA), following the manufacturer’s recommendations. Each kit was confirmed for its sensitivity, recovery, linearity and precision. In the study with the phytochemicals, IL-8 levels were assessed (IL-6 was not secreted from the cells) using an Abcam human IL-8 ELISA kit (ab100575), according to the manufacturer’s protocol.

### 2.10. Quantifying ACE2 Protein

ACE2 protein was quantified using an Abcam human ACE2 ELISA kit (ab235649). Caco-2/TC7 cells were seeded and differentiated in 6-well Transwell plates as mentioned in Section 2.2 and Section 2.3. The cells were treated with the phytochemicals as described in Section 2.7. After treatments, the cells were washed thrice with ice-cold PBS and stored at −80 °C until analysis. On the day of analysis, the cells were lysed in chilled 1X cell extraction buffer PTR supplied by the kit and scraped from the membrane inserts. The lysate was centrifuged at 14,000× *g* for 20 min at 4 °C, and the supernatant was transferred into a freshly labelled tube. ACE2 protein was quantified following the manufacturer’s protocol. The total protein concentration (supernatant) was determined using a Bradford assay, following the manufacturer’s protocol.

### 2.11. Transcriptomics Data Extraction and Processing

To determine whether the correlation observed between *SGLT1* and *TMPRSS2* in Caco-2/TC7 cells applies to human tissues, we performed a search in the NCBI Gene Expression Omnibus (GEO) and the EMBL-EBI ArrayExpress databases for human studies with available transcriptomic data on the small intestine or colon. We examined the association between *SGLT1*, *GLUT2*, *TMPRSS2* and *ACE2* using Pearson correlation analysis in both healthy and inflamed gut samples from five independent transcriptomic datasets encompassing the ileum, colon and rectum. Only studies utilising high-throughput RNA sequencing were included, due to the high accuracy associated with this method. A summary of the included studies is provided in Table 1. For detailed information regarding study design, sample processing and analysis, please refer to the original papers cited in Table 1.

### 2.12. Statistical Analysis

GraphPad Prism 9.0.1 (GraphPad Software, Boston, MA, USA) was used to plot the data and execute all statistical analyses. To compare two independent groups with unequal variances, Welch’s *t*-test was employed. For comparisons between more than two groups, one-way or two-way ANOVA was used, followed by Fisher’s LSD multiple comparison test. The association between *ACE*, *ACE2*, *SGLT1*, *GLUT2* and *TMPRSS2* gene expressions and IL-8 protein was determined using Pearson correlation, where *p* values <0.05 were considered statistically significant.

## 3. Results

### 3.1. Influence of Glucose on the Expression of Target Genes

To test the hypothesis that high glucose induces inflammation in the gut and increases the risk of type 2 diabetes and SARS-CoV-2 viral entry, we first looked at the influence of glucose alone on the expression of *ACE*, *ACE2*, *SGLT1*, *GLUT2* and *TMPRSS2*. As a model, we used the well-characterised human intestinal epithelial cell line Caco-2/TC7. The cells were differentiated for 14 days, at which point expression of *ACE2* and *TMPRSS2* had peaked, as evidenced by our preliminary work (Appendix A). There were no notable effects of acute (4 h) exposure to various sugars (≤50 mM) on the expression of *ACE2* and *TMPRSS2* (Appendix A), and thus the cells were cultured in normal glucose (5.5 mM) or high glucose (25 mM) for the final 7 days of differentiation. The cells did not secrete measurable levels of IL-8, a marker of inflammation, into the basolateral compartment, indicating that high glucose does not induce an inflammatory response, based on IL-8 release, in these cells. However, high glucose alone significantly increased *SGLT1* (*p* < 0.05) and *TMPRSS2* (*p* < 0.01) mRNA expression but not *ACE*, *ACE2* or *GLUT2* (Figure 1).

### 3.2. Effects of IL-1β and TNF-α on IL-8 Secretion, Epithelial Integrity and Cell Viability

The differentiated Caco-2/TC7 cell monolayers were treated with IL-1β (25 ng/mL) and TNF-α (50 ng/mL) for the final 24, 48, 72 or 168 h (Figure 2A), and the inflammatory response was assessed by quantifying the amount of IL-6 and IL-8 in the basolateral compartment. In cells cultured in both normal and high glucose, IL-8 secretion was increased substantially when the cells were exposed to pro-inflammatory cytokines (Figure 2B,C), peaking at 24 h and tailing off to the baseline by 168 h (Figure 2). The cells did not secrete detectable amounts of IL-6. A small but significant drop in transepithelial electrical resistance (TEER) was seen after 24 h and 168 h of cytokine exposure in 5.5 mM glucose-grown cells (Appendix A), while the decline was only evident at 168 h in the 25 mM glucose-grown cells (Appendix A), although the cell monolayers were within an acceptable range (>200 Ω·cm^2^) in all conditions. The cell viability of the cytokine-treated cells was not significantly different from that of the untreated controls (*p* > 0.05) (Appendix A).

### 3.3. Effects of IL-1β and TNF-α on the Expression of Target Genes

Cytokine exposure significantly affected the expression of *ACE*, *ACE2*, *SGLT1*, *GLUT2* and *TMPRSS2* mRNA, genes involved in glucose homeostasis and SARS-CoV-2 infection risk, in Caco-2/TC7 cells. IL-1β and TNF-α increased *ACE2* mRNA in cells cultured in both normal (5.5 mM) and high (25 mM) glucose in a time-dependent manner (Figure 3A,G). *SGLT1* and *TMPRSS2* mRNA also increased in response to the cytokines—both peaked at 24 h in normal glucose and 48 h in high glucose—with *SGLT1* increasing more than all other genes (Figure 3). Notably, *GLUT2* mRNA did not change in normal glucose (Figure 3C) but decreased in high glucose (Figure 3I). Since the changes with time followed different patterns for each gene, we constructed heat maps to visualise the changes (Figure 3F,L) and performed Pearson correlation tests to quantify similarities in the patterns (Figure 3M,N), both of which included the IL-8 data presented in Figure 2. The temporal patterns of *ACE2* expression, for example, were closely matched to *ACE* and IL-8 in high glucose, while the changes in *TMPRSS2* were similar to the changes in *SGLT1*, *GLUT2*, *ACE* and IL-8 in low glucose (Figure 3O). Generally, there were no consistent changes in the expression of the target genes between the cells cultured in 5.5 mM or 25 mM glucose (Appendix A). Thus, moving forward, the cells were cultured in 5.5. mM glucose only.

### 3.4. Initial Testing and Selection of Phytochemicals

In preliminary experiments, we tested a range of phytochemicals, drugs and plant extracts for their capacity to modify the expression of *TMPRSS2* and *ACE2* in Caco-2/TC7 cells, with the justification of their inclusion in the study outlined in Appendix A. Although none modulated *ACE2* mRNA expression, genistein, apigenin, artemisinin and sulforaphane reliably lowered *TMPRSS2* expression (Appendix A). Since changes in mRNA for *TMPRSS2* showed the same temporal pattern and association as glucose transporters, we selected these phytochemicals for further testing in an inflammation model.

Moving forward, the cells were cultured in 5.5 mM glucose for the final 7 days of differentiation and exposed to the cytokine cocktail for the final 72 h, which was deemed optimal for changes in gene expression (Figure 3). Phytochemicals were added for the final 4 h to measure the acute response or twice daily for the final 60 h, starting 12 h after the first addition of the cytokine cocktail, to measure the longer-term effect (Figure 4A).

### 3.5. Effects of Phytochemicals on Inflammation-Induced IL-8 Secretion

None of the tested compounds which decreased *TMPRSS2* mRNA decreased IL-8 secretion after IL-1β and TNF-α stimulation, and at some concentrations, there was an increase in secreted IL-8 protein (Figure 4). Only artemisinin induced further IL-8 secretion after the single 4-h treatment (Figure 4B), while after 60 h, genistein (Figure 4C) and apigenin (Figure 4D) appeared to have a hormetic effect on IL-8 secretion, and artemisinin (Figure 4E) and sulforaphane (Figure 4F) appeared to have a dose-dependent effect, whereby increased concentrations increased IL-8 secretion.

### 3.6. Effects of Phytochemicals on ACE2 mRNA and Protein in Standard and Inflamed Cells

Exposure to IL-1β and TNF-α increased both *ACE2* mRNA (as described above) (Figure 5i) and protein (Appendix A) in the differentiated Caco-2/TC7 monolayers. Acute (4 h) treatment with genistein modestly decreased *ACE2* mRNA expression in both the standard and inflamed cell models, but these changes were not translated to protein. Similarly, modest changes observed in *ACE2* mRNA did not translate into protein following 4 h with apigenin or artemisinin, and sulforaphane did not change the mRNA nor the protein (Figure 5i and Appendix A). In the inflamed model, chronic exposure (60 h) to genistein, apigenin, artemisinin and sulforaphane resulted in a considerable further increase in ACE2 protein, but only minor changes were observed in the cells not treated with cytokines, including a modest increase in ACE2 protein with 25 µM genistein. One notable feature which was particularly apparent when comparing the effects of the compounds in the two models is that the phytochemicals tended to decrease *ACE2* acutely but increase *ACE2* chronically when the cells were inflamed, and the effects were more pronounced after the chronic treatment (Figure 5i and Appendix A).

### 3.7. Effects of Phytochemicals on SGLT1 mRNA in Standard and Inflamed Cells

Acute and chronic genistein treatment decreased *SGLT1* mRNA in both standard and inflamed cell models, alleviating the inflammation-induced increase in *SGLT1* to pre-inflammation levels (≤10 µM, Figure 5ii). Artemisinin and sulforaphane acutely decreased *SGLT1* mRNA in the inflamed model with no effects in the standard model. Chronic apigenin and artemisinin exposure decreased *SGLT1* mRNA in the standard model but increased *SGLT1* under inflammatory conditions. Chronic sulforaphane treatment did not affect the *SGLT1* levels at low concentrations, but at 20 µM, *SGLT1* transcription was upregulated in both models. Similarly, 25 µM genistein enhanced *SGLT1* transcription in the inflamed model, again suggesting a hormetic effect of this compound. Similar to *ACE2*, the phytochemicals tended to decrease *SGLT1* acutely but increase it chronically in the inflamed cells and decrease *SGLT1* chronically in the standard cells (Figure 5ii).

### 3.8. Effects of Phytochemicals on GLUT2 mRNA in Standard and Inflamed Cells

Genistein, apigenin and sulforaphane decreased *GLUT2* mRNA after 4 h and 60 h independent of cytokine exposure. Artemisinin decreased *GLUT2* acutely in inflamed Caco-2/TC7 cells, but no notable changes were seen after 60 h. The phytochemical-induced decrease in *GLUT2* was generally enhanced in the inflamed cells (Figure 5iii).

### 3.9. Effects of Phytochemicals on TMPRSS2 mRNA in Standard and Inflamed Cells

Genistein downregulated *TMPRSS2* in both the standard and inflamed cells after 4 h and 60 h, alleviating the inflammation-induced increase in *TMPRSS2* to pre-inflammation levels (≤10 µM, Figure 5iv) in similar fashion to *ACE2* (Figure 5i) and *SGLT1* (Figure 5ii) at the mRNA level. Chronic apigenin and sulforaphane treatments decreased the *TMPRSS2* mRNA in cells grown under standard conditions but had no noticeable effect in the inflamed cells. In contrast, acute sulforaphane treatment only decreased *TMPRSS2* in the inflamed cells. Artemisinin acutely decreased *TMPRSS2* in the standard and inflamed models, while chronic artemisinin exposure also decreased *TMPRSS2* in the standard model, but at ≤10 µM, it enhanced the cytokine-induced upregulation of *TMPRSS2*. Similar to *ACE2* and *SGLT1*, the phytochemicals tended to decrease *TMPRSS2* acutely but increase it chronically when the cells were inflamed while decreasing *TMPRSS2* chronically in the standard cells (Figure 5iv). A summary of the directional changes in gene expression for the four target genes in response to the compounds is provided in Appendix A.

### 3.10. Association Between ACE2, SGLT1, GLUT2, TMPRSS2 and IL-8 in the Presence of Phytochemicals

There was a clear and substantial correlation (*p* < 0.001) between the expression of *TMPRSS2* and *SGLT1* in both cell models (Figure 6), even in the presence of different phytochemicals. The compounds which changed *TMPRSS2* expression also altered the expression of *SGLT1* in the same direction. Changes in *ACE2* and *GLUT2* were also significantly associated within both cell models. However, an association between *ACE2* and *SGLT1* or *TMPRSS2* was only evident in the inflamed model (Figure 6). The heatmaps summarising the relative changes in each gene also demonstrate the association between *ACE2*, *TMPRSS2*, *SGLT1* and *GLUT2* in the standard (Figure 6C) and inflamed (Figure 6D) cells. No association was seen between IL-8 secretion and our target genes (Figure 6B).

### 3.11. Supporting Information from Transcriptomic Studies

To further investigate whether the associations observed in the Caco-2/TC7 cells are also present in human gut tissues, we analysed transcriptomic data from studies available in the Gene Expression Omnibus (GEO) and EMBL-EBI ArrayExpress databases. When analysing all of the samples as a whole, two out of three studies involving the colon and rectum demonstrated a strong correlation (*r* > 0.5) between *TMPRSS2* and *SGLT1* (Table 1). Conversely, studies involving the ileum exhibited a weak correlation between these two genes (*r* < 0.5) but showed a strong association (*r* > 0.6) between *ACE2* and the glucose transporters *SGLT1* and *GLUT2* (Table 1). A comparison between healthy and inflamed tissues revealed that the correlation between *SGLT1* and *TMPRSS2* increased under inflammatory conditions in most of the studies listed in Table 1.

**Table 1 antioxidants-14-00253-t001:** Pearson correlations between *SGLT1*, *TMPRSS2*, *GLUT2* and *ACE2* mRNA levels in human-derived ileum, colon and rectum tissue samples from patients with and without inflammatory bowel disease.

Study Accession Number	Disease State	Tissue	Sample Category	n	Pearson’s Correlation Coefficient (r)				Citation
					*TMPRSS2* vs. *SGLT1*	*TMPRSS2* vs. *GLUT2*	*ACE2* vs. *SGLT1*	*ACE2* vs. *GLUT2*	
E-MTAB-5783	CD	Ileum	All	68	0.40	0.12	0.69 *	0.64 *	[31]
			Not IBD	32	0.38	0.22	0.66 *	0.68 *	
			CD	36	0.51 *	−0.01	0.56 *	0.38	
GSE57945	CD	Ileum	All	254	0.04	−0.02	0.86 *	0.86 *	[32]
			Not IBD	42	0.55 *	0.67 *	0.79 *	0.78 *	
			CD	174	0.04	−0.04	0.89 *	0.88 *	
			UC	38	0.37	0.14	0.74 *	0.88 *	
GSE174159	CD & UC	Colon	All ^†^	39	0.49	0.05	0.32	−0.07	[33]
			Not IBD	5	0.56 ^#^	0.37	0.44	0.40	
			CD ^†^	17	0.77 *	−0.05	0.46	−0.13	
			CD (quiescent)	11	0.73 *	−0.05	0.45	−0.22	
			CD (moderate)	5	0.77 ^#^	0.15	0.69 ^#^	0.10	
			UC	17	0.38	0.45	0.38	−0.06	
			UC (quiescent)	4	0.50 ^#^	0.25	−0.44	−0.72 ^#^	
			UC (moderate)	9	0.49	0.45	0.44	0.34	
			UC (severe)	4	0.90 ^#^	0.53 ^#^	0.88 ^#^	0.62 ^#^	
GSE117993	CD & UC	Rectum	All	190	0.58 *	−0.04	0.55 *	0.16	[34]
			Not IBD	55	0.45	0.06	0.16	0.00	
			CD	92	0.50 *	−0.07	0.53 *	−0.07	
			UC	43	0.79 *	−0.06	0.38	0.42	
GSE109142	UC	Rectum	All	226	0.72 *	0.08	0.62 *	0.17	[34]
			Not IBD	20	0.59 *	−0.38	0.25	−0.2	
			UC	206	0.75 *	0.20	0.62 *	0.30	
			UC (mild)	53	0.55 *	0.11	0.60 *	0.27	
			UC (moderate to severe)	153	0.80 *	0.22	0.63 *	0.27	

CD = Crohn’s disease; UC = ulcerative colitis; Not IBD = no inflammation control; n = total number of samples; * Strong and statistically significant correlation (*r* > 0.5, *p* < 0.05). ^#^ Strong correlation but not statistically significant due to low sample number (*r* > 0.5, *p* > 0.05). ^†^ Sample GSM5288100 was excluded from analysis due to high deviation from other data.

## 4. Discussion

We investigated the impact of high glucose and pro-inflammatory environments on the expression of ACE2 and glucose transporters in the gut, as well as the potential for phytochemicals to reduce inflammation and glucose transport via ACE2 modulation. Given that ACE2 is highly expressed in the gut and contributes to the pathophysiology of type 2 diabetes [14,15,16], and co-expression of TMPRSS2 with ACE2 is implicated in the pathophysiology of COVID-19 [24,25,26], we also examined the effects of glucose, inflammation and phytochemicals on TMPRSS2. We cultured differentiated human Caco-2/TC7 cells under standard and pro-inflammatory conditions, treating both with phytochemicals shown to downregulate *TMPRSS2* and examining the effects on *ACE2*, *SGLT1*, *GLUT2* and *TMPRSS2* after 4 h (acute) and 60 h (chronic).

### 4.1. High Glucose and Inflammation Increase the Risk of Type 2 Diabetes Through Upregulation of SGLT1

Overexpression of the glucose transporters SGLT1 and GLUT2 may increase intestinal glucose absorption, resulting in postprandial hyperglycaemia, a major risk factor for type 2 diabetes [2,3]. Using Caco-2/TC7 cells as a gut model, we demonstrate the effect of high glucose and inflammation on the expression of *SGLT1* and *GLUT2* in the gut. Prolonged exposure to high glucose (25 mM) and pro-inflammatory cytokines (IL-1β and TNF-α) significantly increased *SGLT1* transcription, with the increase being more profound under inflamed conditions (Figure 1 and Figure 3). Several in vitro and in vivo studies demonstrated increases in SGLT1-mediated glucose transport in response to increasing dietary carbohydrate intake [4,5,6,7] and inflammation [8,9] in gut cells and other tissues [35,36,37]. For example, SGLT1 protein was increased in NCI-H716 cells, another human colorectal cell line, in response to high glucose and cytokines [8]. In the current study, high glucose did not significantly affect *GLUT2* transcription (Figure 1C). However, *GLUT2* was downregulated in Caco-2/TC7 cells in response to inflammation, with the decrease being more profound in the cells differentiated in high glucose (Figure 3I).

The opposing regulatory responses of *SGLT1* and *GLUT2* in the context of inflammation remain unclear but could reflect an evolved defence mechanism by which gut cells protect themselves from the detrimental effects of prolonged inflammation. Such mechanisms may serve to limit excessive glucose transport which could exacerbate metabolic and inflammatory stress. Supporting this notion, intestinal GLUT2 knockdown in mice was found to preserve gut integrity, reduce susceptibility to infection, modulate the gut microbiota composition and reduce systemic inflammation [38,39]. Moreover, evidence from a mouse model of Listeria infection demonstrated that SGLT1 knockout not only increased bacterial spread but also led to the death of SGLT1-deficient mice [40], emphasising the critical yet distinct roles of these glucose transporters in maintaining gut homeostasis and systemic health.

### 4.2. High Glucose and Inflammation Increase the Risk of COVID-19 by Upregulating ACE2 and TMPRSS2 in the Gut

Given the importance of intestinal ACE2 in regulating inflammation and glucose transport [15,16,41], we hypothesised that changes in ACE2 and TMPRSS2 due to low-grade inflammation and hyperglycaemia potentially contribute to severe disease development in COVID-19 patients with diabetes. Our preliminary work showed that *ACE2* and *TMPRSS2* were abundantly expressed in Caco-2/TC7 enterocytes, increasing with differentiation but stabilising after 14 days (Appendix A). Therefore, we examined how high glucose and inflammation, key features of diabetes, affect their expression in 14-day differentiated cells. When the cells were exposed to various concentrations of glucose, fructose and sucrose (≤50 mM) for 4 h, there were no consistent changes in *ACE2* or *TMPRSS2* (Appendix A). However, prolonged exposure (7 days) to high glucose (25 mM) increased *TMPRSS2* by 32% compared with the cells cultured in normal (5.5 mM) glucose, with no change in *ACE2* (Figure 1). High glucose also triggered an inflammatory response, as evidenced by higher amounts of IL-8 secreted into the basolateral compartment (Figure 2). Additionally, exposure to pro-inflammatory cytokines increased both *ACE2* and *TMPRSS2* under normal and high-glucose conditions (Figure 3 and Appendix A). These findings suggest that high glucose and inflammation upregulate ACE2 and TMPRSS2, potentially facilitating viral replication and spread, which may worsen COVID-19 severity in patients with diabetes and other inflammatory disorders if the same effects are translated into other organs.

### 4.3. Association Between ACE2, TMPRSS2, Glucose Transporters and Inflammation: A Mechanism for the Increased Risk of Type 2 Diabetes with COVID-19?

We observed a significant inverse association between *ACE2* mRNA and secreted IL-8 protein; while *ACE2* transcription increased with the length of cytokine treatment, IL-8 levels continued to fall sharply. Similarly, patients with inflammatory bowel disease responding to treatment showed increased expression of *ACE2* [42], pointing towards an anti-inflammatory function from ACE2 in the gut. ACE2 is an important regulator of gut inflammation [41]. Lowered *ACE2* expression in the small intestine was associated with inflammation, disease relapse, non-responsiveness to therapy and the emergence of complex disease in patients with Crohn’s [42,43]. In addition to *ACE2*, expression of sodium-dependent neutral amino acid transporter (*B^0^AT1*) was also decreased in the small intestine of those with active Crohn’s disease [43]. *B^0^AT1* transcription in Caco-2/TC7 cells was examined, but the data are not shown due to extremely low expression (<5 copies per ng cDNA).

As well as regulating gut inflammation, recent studies demonstrated a link between ACE2 and the glucose transporters SGLT1 and GLUT2 [15,16]. ACE2-generated Ang-(1-7) binding to the Mas receptor blocks glucose absorption in the gut by regulating SGLT1 and GLUT2 expression (Figure 7). However, as shown by Pearson’s correlation analysis, we found no significant correlation between the glucose transporters and *ACE2* when the cells were exposed to pro-inflammatory cytokines (Figure 3). In contrast, a strong correlation between *ACE2* and the glucose transporters was observed in patient-derived ileum samples (Table 1), suggesting a tissue-specific role in gene expression. Interestingly, changes in the expression of both *SGLT1* (*p* < 0.001) and *GLUT2* (*p* < 0.01) were significantly associated with changes in *TMPRSS2* expression in Caco-2/TC7 cells (Figure 3). Similarly, a substantial correlation between *SGLT1* and *TMPRSS2* was observed in the patient-derived colon and rectum samples (Table 1) but not in the samples derived from the ileum (Table 1), reinforcing the idea of a tissue-specific role in gene association.

Similar to our findings, studies have shown an inflammation-induced increase in *TMPRSS2* expression in the gut and other tissues [44,45,46]. In comparison with wild-type mice, SARS-CoV and MERS-CoV replication in the lungs was low in *TMPRSS2* knock-out mice, as was the virus-induced pro-inflammatory response, particularly the production of cytokines and chemokines [47]. Suárez-Fariñas et al. identified that *TMPRSS2* in the gut was colocalised with gene products associated with basic epithelial function, including tight junction formation [42]. Activation of matriptase, a proposed substrate of TMPRSS2 [48], protected mice from dextran sodium sulfate-induced experimental colitis and promoted intestinal barrier function [49]. Further, along with *ACE2*, *TMPRSS2* also increased in patients with inflammatory bowel disease responding to IL-12- and IL-23-targeted treatments [42]. It remains to be seen if the role of TMPRSS2 is protective or pro-inflammatory, and further studies are required to elucidate the association between the glucose transporters and TMPRSS2.

*SGLT1* was strongly positively correlated with IL-8 (Figure 3), indicating that inflammation potentially increases glucose absorption in the gut, as seen previously [8]. In addition to diabetes being a risk factor for COVID-19, evidence suggests that COVID-19 triggers the development of new-onset diabetes. The risk of developing diabetes is 40% higher in people who have had COVID-19 than those without [50]. Various hypotheses have been developed to describe the role of contracting COVID-19 in new-onset diabetes, including dysregulated immune system, persistent low-grade inflammation, pancreatic β-cell dysfunction, COVID-19 treatment-induced side effects (use of glucocorticoids and antibiotics), stress and autoimmune mechanisms [50,51]. Further, COVID-19 patients are 36% more likely than non-COVID-19 patients to develop long-term gastrointestinal issues [52]. Even seven months after infection, most individuals with long COVID and inflammatory bowel disease had SARS-CoV-2 antigens in their gut mucosa, potentially contributing to immune perturbations [53]. Given this, we propose that the increased risk of type 2 diabetes post-COVID-19 is partly related to inflammation-induced upregulation of the glucose transporter SGLT1 in the gut, which could contribute to postprandial hyperglycaemic spikes, insulin resistance and eventually development of type 2 diabetes.

### 4.4. Genistein Downregulates GLUT2 and Alleviates Inflammation-Induced Increases in SGLT1 and TMPRSS2

Genistein, apigenin, artemisinin and sulforaphane all increased ACE2 protein expression in the inflamed cell model (Appendix A). However, none reduced IL-8 secretion, with some even promoting IL-8 production at high concentrations (Figure 4), likely due to their pro-oxidant effects at high doses [54,55]. Despite this, genistein was the most promising of the four compounds, effectively and reliably alleviating inflammation-induced increases in *SGLT1* and *TMPRSS2* mRNA and restoring them to pre-inflammation levels (Figure 5ii,iv). *GLUT2* was also significantly decreased by genistein in both cell models. Since genistein lowers *SGLT1* and *GLUT2* mRNA, consuming genistein-containing foods may aid both healthy individuals and those with persistent inflammation, such as those with metabolic disorders and inflammatory gastrointestinal disorders, in lowering postprandial glycaemic response (Figure 7). There is growing evidence that TMPRSS2 inhibitors can effectively block SARS-CoV-2 infection [47,56,57], and studies suggest that TMPRSS2 may be more crucial in SARS-CoV-2 infection than ACE2 [58], while maintaining ACE2 alleviates inflammation and glucose transport [15,16,41]. Therefore, reduced *TMPRSS2* but increased *ACE2* in response to chronic exposure (60 h) to genistein in inflamed Caco-2/TC7 cells could be considered beneficial in attenuating the risk of SARS-CoV-2 infection and COVID-19 pathophysiology (Figure 7).

The ability of genistein to combat inflammation-induced changes in gene expression is likely linked to its potent dual action in neutralising free radicals and modulating redox-sensitive signalling pathways [59,60,61]. IL-1β and TNF-α activate downstream signalling pathways, such as nuclear factor kappa-light-chain-enhancer of activated B cells (NF-κB), activated protein 1 (AP-1) and mitogen-activated protein kinase (MAPK), leading to upregulation of reactive oxygen species (ROS)-producing enzymes like inducible nitric oxide synthase and NADPH oxidases [62,63]. This intensifies oxidative damage and drives the expression of proinflammatory genes, exacerbating inflammation. Genistein inhibits the activation of NF-κB and MAPK pathways, reducing the nuclear translocation of transcription factors such as NF-κB and AP-1 [64,65]. Concurrently, genistein activates the nuclear factor erythroid 2-related factor 2 (Nrf2) pathway, enhancing the transcription of antioxidant enzymes such as superoxide dismutase, catalase and glutathione peroxidase, further mitigating oxidative stress and curtailing downstream inflammatory responses [59,60].

### 4.5. Association Between SGLT1 and TMPRSS2: A Common Transcription Factor Implicated in Pathological Conditions?

There was a strong, significant correlation (*p* < 0.001) between *TMPRSS2* and *SGLT1* (Figure 6) independent of phytochemical treatment, further suggesting that a common transcriptional factor may regulate these two genes. Additionally, this association increased during inflammation (Figure 6), a finding consistent across patient-derived samples (Table 1). Regardless of tissue origin, the correlation between *TMPRSS2* and *SGLT1* increased in most samples during inflammation (Table 1). For instance, in studies GSE174159 and GSE109142, the association between *TMPRSS2* and *SGLT1* clearly intensified with disease severity (Table 1). Given that this association strengthened with disease severity, coupled with the fact that Caco-2/TC7 cells are a cancer-derived cell line, it can be postulated that there is a potential association between SGLT1 and TMPRSS2 in pathological conditions.

### 4.6. Study Limitations

In our study, the effect of genistein on *SGLT1*, *GLUT2* and *TMPRSS2* expression raises the possibility that it could be a promising agent in reducing postprandial hyperglycaemia and SAR-CoV-2 infection risk. However, several limitations need to be considered. One major limitation is that we only measured *SGLT1*, *GLUT2* and *TMPRSS2* mRNA, as mRNA does not always reflect protein levels. Phytochemicals, in addition to affecting gene and protein expression at the molecular level, can impede the function of these receptors by binding to them. Another limitation is the use of Caco-2/TC7 cells instead of human primary cells. The neutral amino acid transporter *B^0^AT1* was extremely lowly expressed in Caco-2/TC7 cells, which limited our ability to monitor the association between *B^0^AT1*, inflammation and *ACE2*, despite many studies showing a link between them [16,41]. While it is unclear how much the human colon-derived cancer cell line would represent the situation in vivo, it is noteworthy that the changes observed in *ACE2* and *TMPRSS2* in ulcerative colitis patients matched the changes we observed in inflamed Caco-2/TC7 cells [43], suggesting the potential of Caco-2/TC7 cells as a preliminary model to assess gut-related functions in vitro.

## 5. Conclusions

High glucose and inflammation upregulate *ACE2*, *SGLT1* and *TMPRSS2* in Caco-2/TC7 cells, potentially increasing the risk of type 2 diabetes and SARS-CoV-2 infection in humans. These findings suggest a possible relationship between elevated SGLT1 and new-onset diabetes in COVID-19 patients and may help to explain why individuals with pro-inflammatory diseases, like diabetes and obesity, are more likely to develop severe COVID-19, which we propose occurs via a gut-mediated mechanism. Furthermore, our results provide mechanistic evidence suggesting that genistein may be utilised as a preventive measure against the inflammation-induced rise in *SGLT1* and *TMPRSS2*, potentially aiding in lowering postprandial glycaemic response and COVID-19 pathophysiology. The observed strong correlation between *TMPRSS2* and *SGLT1* expression indicates that both genes may be regulated by a common yet unidentified transcriptional factor or signalling pathway and are likely upregulated in response to inflammatory conditions. The association between *ACE2* and the glucose transporters appears to be context-dependent, with no evidence of direct gene regulation seen. Further research is required to assess the importance of genistein in vivo and to study the link between SGLT1 and TMPRSS2.

## Figures and Tables

**Figure 1 antioxidants-14-00253-f001:**
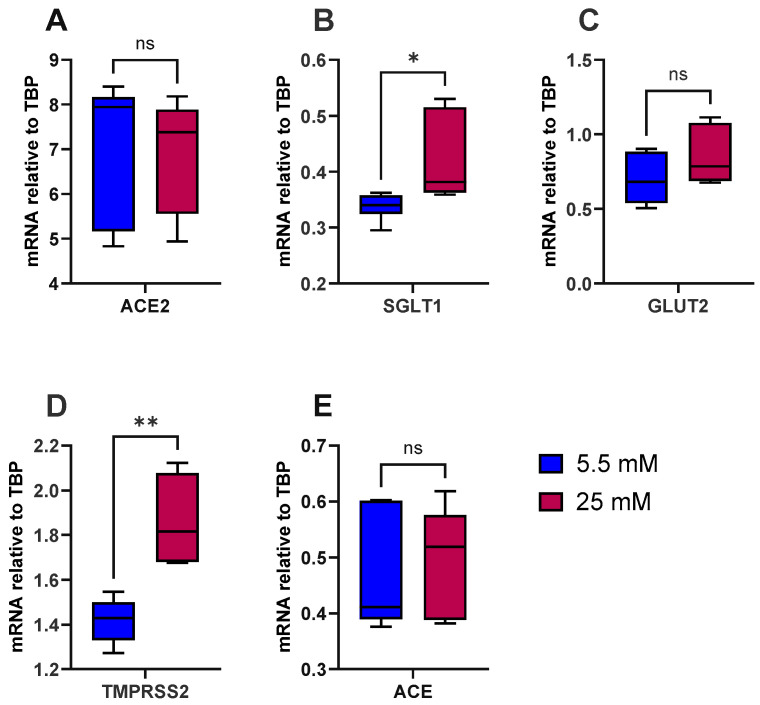
Effect of glucose concentration on gene expression in Caco-2/TC7 cells. After 7 days of differentiation, Caco-2/TC7 cells were cultured in medium containing either 5.5 mM glucose (normal glucose) or 25 mM glucose (high glucose) for the next 7 days. Total RNA was extracted from the cells after 14 days of differentiation and reverse transcribed (RT) to synthesise cDNA, which was partitioned, amplified and analysed via ddPCR. Gene expression was quantified as absolute copies per ng mRNA, assuming a 1:1 RT efficiency, and (**A**) *ACE2*, (**B**) *SGLT1*, (**C**) *GLUT2*, (**D**) *TMPRSS2* and (**E**) *ACE* are presented relative to the *TBP* reference gene. Data were collected independently from cells across three biological passages, analysed in duplicate and presented as the mean ± SD (n/N = 6/3). Significant differences were determined using an unpaired *t*-test with Welch’s correction (ns = not significant). * *p* < 0.05. ** *p* < 0.01.

**Figure 2 antioxidants-14-00253-f002:**
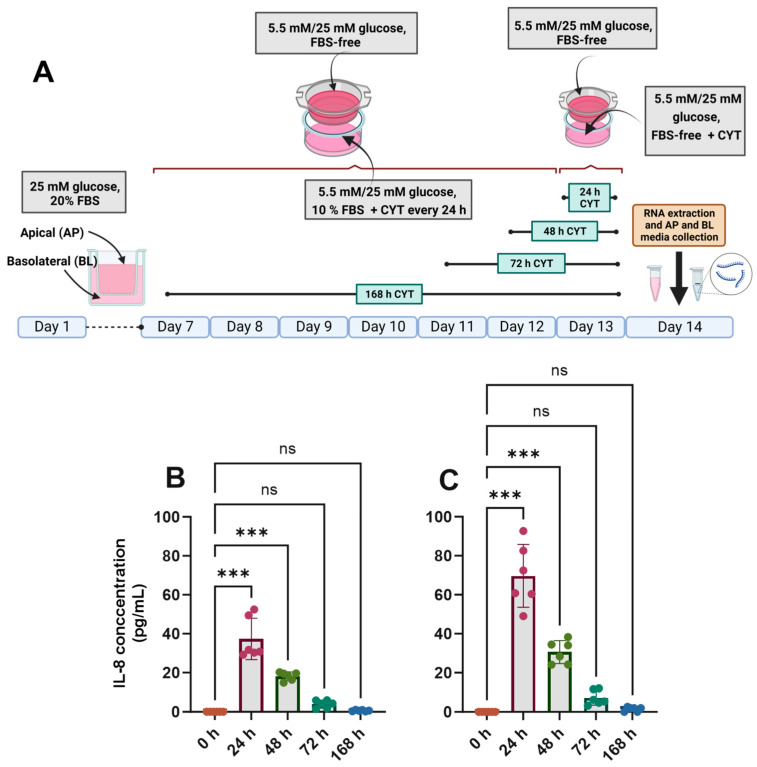
Effect of glucose concentration on IL-8 secretion in cytokine-treated Caco-2/TC7 cells. (**A**) Cells were cultured in 25 mM glucose for the first 7 days of differentiation and in either 5.5 mM (normal glucose) or 25 mM (high glucose) glucose until day 14. A cytokine cocktail containing IL-1β (25 ng/mL) and TNF-α (50 ng/mL) was added once per day to the basolateral compartment to induce inflammation daily for ≤168 h during the second 7 days of differentiation. At the end of day 14, culture media from both compartments (apical and basolateral) were collected for IL-6 and IL-8 estimation, while the total RNA was extracted from the cells for gene expression studies. The culture media from three biological passages were processed, and IL-6 and IL-8 were measured in duplicate using multiplex ELISA. IL-6 was not detected, but IL-8 was secreted into the basolateral compartment in cells cultured in (**B**) 5.5 mM glucose and (**C**) 25 mM glucose, and they are presented as the mean ± SD (n/N = 6/3). AP = apical compartment; BL = basolateral compartment; CYT = cytokine cocktail (IL-1β and TNF-α); FBS = foetal bovine serum. Significant differences were determined using one-way ANOVA with Fisher’s LSD multiple comparisons test (ns = not significant). *** *p* < 0.001. Created with (**A**) BioRender.com (https://BioRender.com/b24a565, accessed 14 January 2025) and (**B**,**C**) GraphPad Prism version 9.0.1.

**Figure 3 antioxidants-14-00253-f003:**
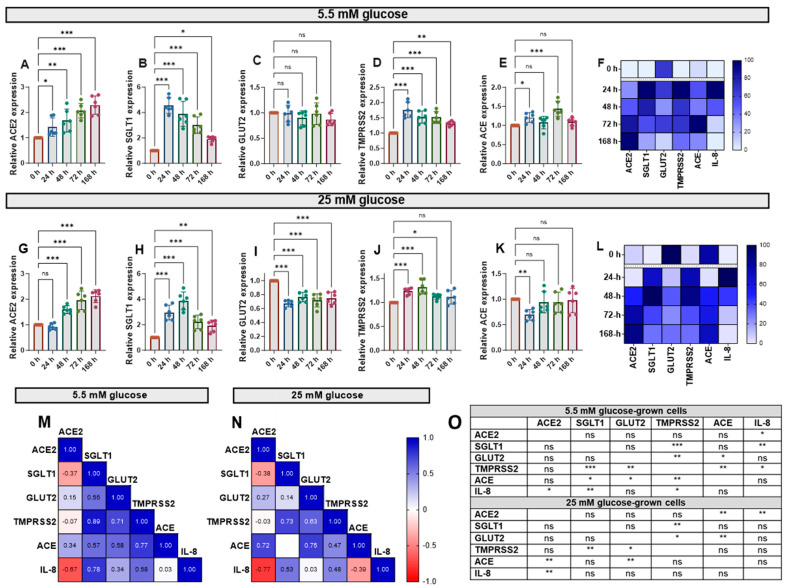
Effect of inflammation on *ACE2*, *SGLT1*, *GLUT2*, *TMPRSS2* and *ACE* mRNA in Caco-2/TC7 cells cultured in normal or high glucose. Cells were cultured in 25 mM glucose for the first 7 days of differentiation and in either 5.5 mM (normal glucose) or 25 mM (high glucose) glucose until day 14. A cytokine cocktail containing IL-1β (25 ng/mL) and TNF-α (50 ng/mL) was added to the basolateral compartment to induce inflammation for ≤168 h during the second 7 days. At the end of day 14, culture media from both compartments (apical and basolateral) were collected for IL-6 and IL-8 estimation, while the total RNA was extracted from cells for gene expression studies (Figure 2A). The total RNA was reverse transcribed (RT) to synthesise cDNA, which was partitioned, amplified and analysed via ddPCR. Gene expression was quantified as absolute copies per ng mRNA relative to *TBP*, assuming a 1:1 RT efficiency, and presented as the mean fold change compared with the control without cytokine exposure (0 h) across three biological passages assayed in duplicate. Changes in mRNA expression following cytokine exposure in cells grown in 5.5 mM glucose for (**A**) *ACE2*, (**B**) *SGLT1*, (**C**) *GLUT2*, (**D**) *TMPRSS2* and (**E**) *ACE* and in 25 mM glucose for (**G**) *ACE2*, (**H**) *SGLT1*, (**I**) *GLUT2*, (**J**) *TMPRSS2* and (**K**) *ACE*. Data are expressed as the mean ± SD (n/N = 6/3). Heatmaps summarising the relative changes in the target genes and IL-8 secretion in response to the pro-inflammatory environment in cells cultured in (**F**) 5.5 mM and (**L**) 25 mM glucose are shown. Data are normalised relative to the smallest and largest value in each dataset, with 0 and 100 representing the lowest and largest relative expression value of any of the genes in each dataset, respectively (N = 15). Association between *ACE2*, *SGLT1*, *GLUT2*, *TMPRSS2* and *ACE* gene expression and IL-8 secretion in cells cultured in (**M**) 5.5 mM and (**N**) 25 mM glucose (N = 12). Pearson’s correlation coefficient (r) values are shown in the cells. Blue represents a positive correlation, and red represents a negative correlation. (**O**) The table depicts the significance of the relationship between the examined parameters. Significant differences were determined by one-way ANOVA with Fisher’s LSD multiple comparisons test (ns = not significant). * *p* < 0.05. ** *p* < 0.01. *** *p* < 0.001.

**Figure 4 antioxidants-14-00253-f004:**
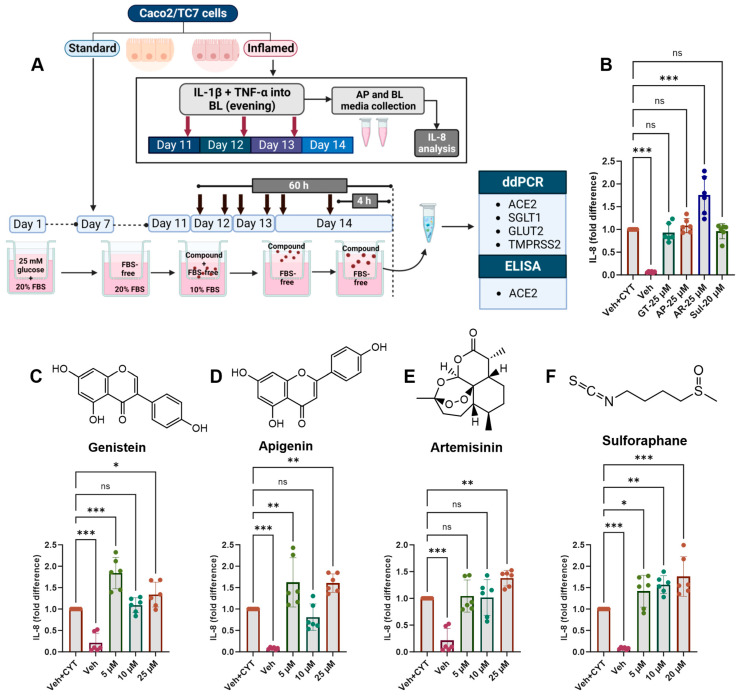
Acute and chronic effects of phytochemicals on IL-8 secretion in inflamed Caco-2/TC7 cells. (**A**) Caco-2/TC7 cells were cultured in 25 mM glucose for the first 7 days of differentiation and then in 5.5 mM glucose until day 14. A phytochemical or 0.1% (*v*/*v*) DMSO vehicle control (Veh) was added to the apical compartment twice daily for the final 60 h or once for the final 4 h. In the inflamed model, a cytokine cocktail containing IL-1β (25 ng/mL) and TNF-α (50 ng/mL) (CYT) was added to the basolateral compartment once daily for the final 72 h. At the end of the treatment regimen, the culture media from the basolateral compartments in the inflamed cell model were collected for IL-8 estimation, while the total RNA or protein was extracted from the cells in both the standard and inflamed models to measure gene expression via ddPCR (*ACE2*, *SGLT1*, *GLUT2* and *TMPRSS2*) or ACE2 protein via ELISA, respectively. The culture media from three biological passages were processed, and IL-8 was measured in duplicate by ELISA, corrected for the total protein, and presented as the mean fold changes compared with the inflamed controls with DMSO as a vehicle control (Veh + CYT) following (**B**) the single 4-h phytochemical treatment or the 60 h of exposure to (**C**) genistein, (**D**) apigenin, (**E**) artemisinin or (**F**) sulforaphane. Data are presented as the mean ± SD (n/N = 6/3). AP = apical compartment; BL = basolateral compartment; FBS = foetal bovine serum; GT = genistein; AP = apigenin; AR = artemisinin; Sul = sulforaphane. Significant differences were determined by one-way ANOVA with Fisher’s LSD multiple comparisons test (ns = not significant). * *p* < 0.05. ** *p* < 0.01. *** *p* < 0.001. Created with (**A**) Biorender.com (https://BioRender.com/k84t496, accessed 14 January 2025) and (**B**–**F**) GraphPad Prism version 9.0.1.

**Figure 5 antioxidants-14-00253-f005:**
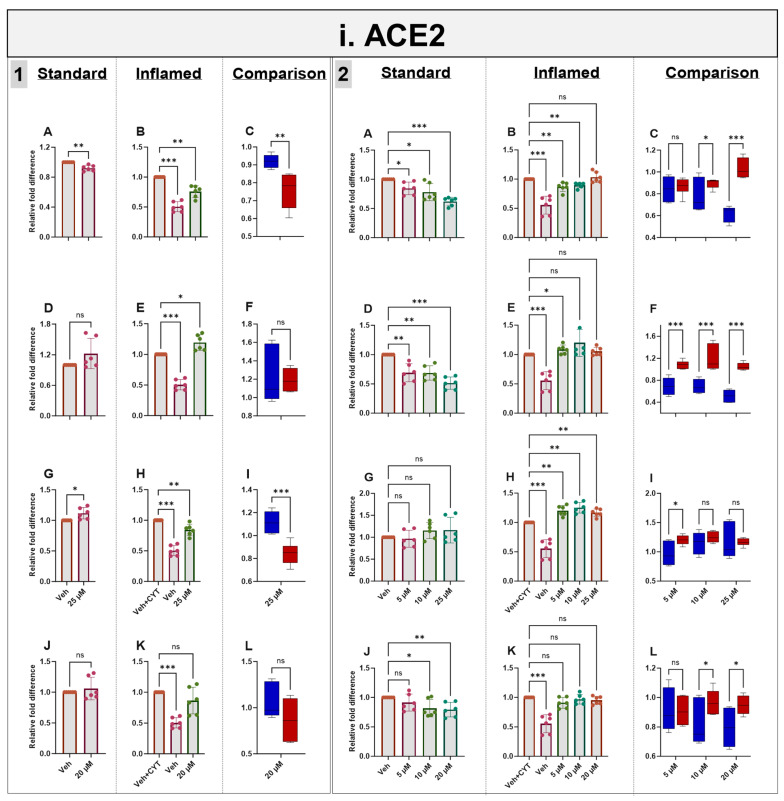
Acute and chronic effects of phytochemicals on *ACE2*, *SGLT1*, *GLUT2* and *TMPRSS2* mRNA in standard and inflamed Caco-2/TC7 cells. The acute (**Panel 1**) and chronic (**Panel 2**) effects of genistein (**A**–**C**), apigenin (**D**–**F**), artemisinin (**G**–**I**) and sulforaphane (**J**–**L**) on (**i**) *ACE2*, (**ii**) *SGLT1*, (**iii**) *GLUT2* and (**iv**) *TMPRSS2* mRNA were measured in Caco-2/TC7 cells cultured in standard (first column of each panel) or inflammatory (second column of each panel) conditions. Cells were treated with the phytochemicals (5, 10 or 25 µM genistein, apigenin or artemisinin or 5, 10 or 20 µM sulforaphane) or 0.1% (*v*/*v*) DMSO vehicle control (Veh) for either 4 h (acute) or 60 h (chronic) with or without exposure to the cytokine cocktail of IL-1β (25 ng/mL) and TNF-α (50 ng/mL) for 72 h (+CYT), as outlined in Figure 4A. A comparison of the effect of each compound in the standard (blue) and inflammatory (red) conditions is shown in the third column of each panel. For gene expression, the total RNA was extracted and reverse transcribed (RT) to synthesise cDNA, which was partitioned, amplified and analysed via ddPCR. Gene expression was quantified as absolute copies per ng mRNA relative to *TBP*, assuming a 1:1 RT efficiency, and presented as the mean fold change compared with the control (with or without cytokine exposure accordingly) across three biological passages assayed in duplicate. Data are expressed as the mean ± SD (n/N = 6/3). Significant differences were determined as follows. Panel 1: Standard and comparison using unpaired *t*-test with Welch’s correction and inflamed using one-way ANOVA with Fisher’s LSD multiple comparisons test. Panel 2: Standard and inflamed using one-way ANOVA with Fisher’s LSD multiple comparisons test and comparison using two-way ANOVA with Fisher’s LSD multiple comparisons test (ns = not significant). * *p* < 0.05. ** *p* < 0.01. *** *p* < 0.001.

**Figure 6 antioxidants-14-00253-f006:**
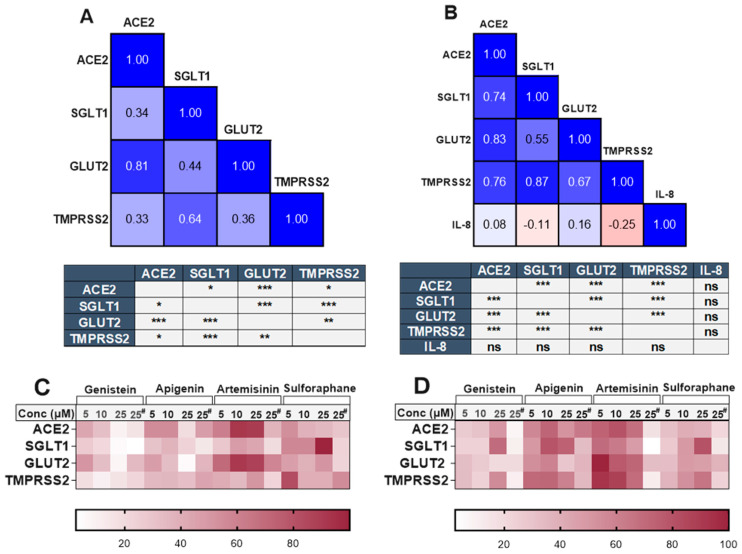
Association between *ACE2*, *SGLT1*, *GLUT2*, *TMPRSS2* and IL-8 in standard and inflamed Caco-2/TC7 cells exposed to phytochemicals. Pearson’s correlation matrix showing the association between *ACE2*, *SGLT1*, *GLUT2* and *TMPRSS2* gene expression and the tables (below the matrix) depicting the significance of the relationships between the examined parameters in (**A**) standard Caco-2/TC7 and between *ACE2*, *SGLT1*, *GLUT2* and *TMPRSS2* gene expression and IL-8 secretion in (**B**) inflamed Caco-2/TC7 cells, all treated with genistein, apigenin, artemisinin, sulforaphane or a 0.1% (*v*/*v*) DMSO vehicle control for 4 h or 60 h, as outlined in Figure 4A (N = 55). Pearson’s correlation coefficient (*r*) values are shown in the cells. Blue represents a positive correlation, and red represents a negative correlation. Heatmaps summarising the relative changes in the target genes by all of the compounds in the (**C**) standard and (**D**) inflamed cells are shown. Each row denotes a gene, and the columns represent different concentrations of the phytochemicals (^#^ 4-h treatment time, while the rest are 60 h). Data were normalised relative to the smallest and largest value in each row, with 0 and 100 representing the smallest and largest relative expression values of the genes in each dataset, respectively (N = 48), where ns = not significant. * *p* < 0.05. ** *p* < 0.01. *** *p* < 0.001.

**Figure 7 antioxidants-14-00253-f007:**
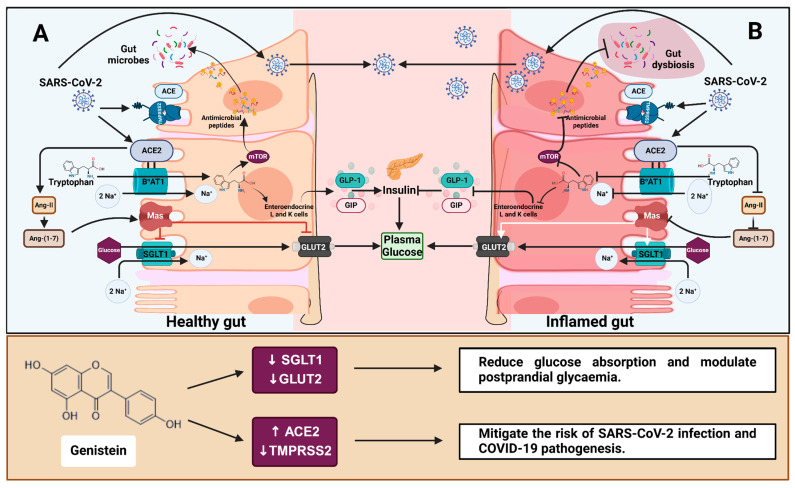
The role of ACE2 in gut health, glucose homeostasis and COVID-19 and the effect of genistein. (**A**) ACE2 and the amino acid transporter B^0^AT1 mediate tryptophan absorption in a healthy gut epithelium. Basolateral tryptophan promotes GLP-1 and GIP secretion in enteroendocrine L- and K-cells. These incretins protect tight junctions from gut dysbiosis and activate pancreatic β cells while suppressing α cells, all of which help to regulate plasma glucose levels. Furthermore, the binding of ACE2-generated Ang-(1-7) to the Mas receptor inhibits glucose absorption via SGLT1 and GLUT2. (**B**) During inflammation, tryptophan absorption and Ang-(1-7) production are impaired, resulting in gut dysbiosis and increased glucose transport in circulation [16]. ACE2 and TMPRSS2 facilitate SARS-CoV-2 entry into host cells, which causes an inflammatory response and contributes to gut dysbiosis and dysfunction. Genistein reduces *SGLT1* and *GLUT2* mRNA, potentially decreasing intestinal glucose absorption and the risk of type 2 diabetes. Additionally, genistein increases ACE2 protein and decreases *TMPRSS2* mRNA, potentially mitigating the risk of SARS-CoV-2 infection and COVID-19 pathogenesis. Regular arrows in the top box (**A**,**B**) indicate transport or secretion, and blunt arrows indicate inhibition. ACE = angiotensin-converting enzyme; ACE2 = angiotensin-converting enzyme 2; Ang II = angiotensin 2; Ang-(1-7) = angiotensin-(1-7); B^0^AT1 = sodium-dependent neutral amino acid transporter; GLUT2 = glucose transporter 2; Mas = Mas receptor; mTOR = mechanistic target of rapamycin; SGLT1 = sodium-dependent glucose cotransporter 1; GLP-1 = glucagon-like peptide 1; GIP = glucose-dependent insulinotropic polypeptide; TMPRSS2 = transmembrane protease, serine 2. Created with BioRender.com (https://BioRender.com/e74a285, accessed 14 January 2025).

## Data Availability

The raw data supporting the conclusions of this article will be made available by the authors on request.

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
