# Peer review of "Impact of Glucose, Inflammation and Phytochemicals on ACE2, TMPRSS2 and Glucose Transporter Gene Expression in Human Intestinal Cells"

_antioxidants, 2025, doi:10.3390/antiox14030253_

Round 1
Reviewer 1 Report
This is a paper addressing the effects of the exposure of human intestinal ephitelial cells (CACO-2/TC7) to glucose and pro-inflammatory agents. Results are mostly based on the expression and regulation of ACE2, TMPRSS2, SGLT1, and GLUT2. In addition, the effects of certain phytochemicals were studied, showing protection in same of the parameters evaluated. Although some of the results are of interest, the manner in which the work is introduced and discussed reduces significantly the enthusiasm for its publication.
My major critics are:
1) mixing mechanistically inflammation, intestinal permeability, diabetes and viral infection is really a difficult task. COVID-19 associations are rather secondary for the experiment performed. Why not just include it in one paragraph of the Discussion?
2) it is unclear how the four phytochemicals used were selected; in addition the asymmetrical characteristics of the 4 molecules does not allow molecular comparisons. Finally, why only genistein is discussed?
3) The use of ‘antioxidant’ to name phytochemicals is rather unnecessary. Is artemisinin an antioxidant? Can it scavenge a free radical? Sulphoraphane and Nrf2? All the discussion on the redox possibilities of genistein (lines 628-638) is rather speculative and no experiments are performed in that direction.
4) As mentioned the Discussion is long, and complicated mixing facts, with potential mechanisms, and potential relevance for human health. Just an example in line 490: “and contribute to type 2 diabetes in COVID-19 patients...” is this a corroborated fact?? Try to reduce the Discussion.
none
Author Response
Reviewer 1
This is a paper addressing the effects of the exposure of human intestinal epithelial cells (CACO-2/TC7) to glucose and pro-inflammatory agents. Results are mostly based on the expression and regulation of ACE2, TMPRSS2, SGLT1, and GLUT2. In addition, the effects of certain phytochemicals were studied, showing protection in same of the parameters evaluated. Although some of the results are of interest, the manner in which the work is introduced and discussed reduces significantly the enthusiasm for its publication.
My major critics are:
Comment 1: Mixing mechanistically inflammation, intestinal permeability, diabetes and viral infection is really a difficult task. COVID-19 associations are rather secondary for the experiment performed. Why not just include it in one paragraph of the Discussion?
Response 1: As suggested, we have revised the manuscript by streamlining the discussion on COVID-19 and shortening its coverage. Instead, we have placed greater emphasis on the interaction between ACE2 and glucose transporters, making this the focal point of our discussion.
Comment 2: It is unclear how the four phytochemicals used were selected; in addition the asymmetrical characteristics of the 4 molecules does not allow molecular comparisons. Finally, why only genistein is discussed?
Response 2: The rationale for the selection of the four compounds was already presented in the text (now lines 323-330) and in Supplementary Table 1 – a range of compounds were screened for effects on expression of ACE2 and TMPRSS2 in Caco-2/TC7 cells, with the four that were investigated further being the most promising. Of these four, genistein was the most promising because it effectively and reliably alleviated inflammation-induced increases in SGLT1 and TMPRSS2 mRNA. We have made this clearer in the Results (lines 323-330) and Discussion (lines 547-548 and lines 552-553).
Comment 3: The use of ‘antioxidant’ to name phytochemicals is rather unnecessary. Is artemisinin an antioxidant? Can it scavenge a free radical? Sulphoraphane and Nrf2? All the discussion on the redox possibilities of genistein (lines 628-638) is rather speculative and no experiments are performed in that direction.
Response 3: We agree that the use of ‘antioxidant’ is unnecessary when referring to phytochemicals but the managing editor requested this approach after our initial submission. We have therefore decided to remove ‘antioxidant’ and refer to the compounds simply as phytochemicals throughout. However, in keeping with the editor’s request to include more discussion around antioxidant properties of the compounds and because we feel that our discussion around the redox properties of genistein (lines 567-579) is a valid potential mechanism of its action on gene expression, as per the cited literature, we have retained this part of the Discussion.
Comment 4: As mentioned the Discussion is long, and complicated mixing facts, with potential mechanisms, and potential relevance for human health. Just an example in line 490: “and contribute to type 2 diabetes in COVID-19 patients...” is this a corroborated fact?? Try to reduce the Discussion.
Response 4: We have shortened the Discussion section by removing redundant or speculative statements and refining the focus of our discussion on our key findings. This includes rephrasing the above sentence to, “Given that ACE2 is highly expressed in the gut and contributes to the pathophysiology of type 2 diabetes, and co-expression of TMPRSS2 with ACE2 is implicated in the pathophysiology of COVID-19…” and we have cited literature that corroborates this (lines 430-432).
Comment 5: [The introduction] needs to be focused to the experiments and results.
Response 5: The entire Introduction has been revised to ensure a clearer focus on the background and rationale for our study, emphasising the key research questions and hypotheses that directly relate to our findings.
Reviewer 2 Report
The manuscript by Rizliya Visvanathan et al., entitled ‘Impact of glucose, inflammation and phytochemical antioxidants on ACE2, TMPRSS2 and glucose transporter gene expression in human intestinal cells’
I have concerns about the following:
- The study primarily focuses on investigating the role of high glucose environment and inflammation on ACE2 and glucose transporter expression. However, it does not explore how phytochemical antioxidants could be used to attenuate inflammation and glucose transport by modulating the ACE2/Ang-(1-7)/Mas receptor axis in standard (non-inflamed) and inflamed Caco-2/TC7 cell models, as stated in the introduction.
- Also, the authors should specify the cell passages used in the experiments and provide a rationale for their selection.
- The manuscript lacks a comprehensive discussion of the interplay between multiple factors acting on inflammation and glucose transport.
- The authors do not identify the limitations of their work.
- Finally, the authors have to clarify the down-regulation of GLUT2 in response to inflammation, which is at odds with the original hypothesis.
It is indicated in the comments to the authors
Author Response
Comment 1: The study primarily focuses on investigating the role of high glucose environment and inflammation on ACE2 and glucose transporter expression. However, it does not explore how phytochemical antioxidants could be used to attenuate inflammation and glucose transport by modulating the ACE2/Ang-(1-7)/Mas receptor axis in standard (non-inflamed) and inflamed Caco-2/TC7 cell models, as stated in the introduction.
Response 1: The Introduction has been revised to ensure that the objectives of the study accurately reflect the experiments performed.
Comment 2: Also, the authors should specify the cell passages used in the experiments and provide a rationale for their selection.
Response 2: We have included this in the Methods section (lines123-124).
Comment 3: The manuscript lacks a comprehensive discussion of the interplay between multiple factors acting on inflammation and glucose transport. Explain the mechanistic aspects of the proposed phenomena in the Introduction.
Response 3: The Introduction and Discussion sections have been revised to better explain the mechanistic aspects of the proposed phenomena, ensuring a clearer link between inflammation, ACE2 regulation and expression of glucose transporters.
Comment 4: The authors do not identify the limitations of their work.
Response 4: A section outlining key limitations has been added to the Discussion (lines 595-610).
Comment 5: Finally, the authors have to clarify the down-regulation of GLUT2 in response to inflammation, which is at odds with the original hypothesis.
Response 5: A paragraph clarifying this has been added to the Discussion (lines 454-464) to explain this observation.
Reviewer 3 Report
Whereas the data are clear, the overall presentation and organization of the paper deserves the interest of the study. The paper therefore needs major modifications.
* Title must be reconsidered
* Abstract must be reconsidered.
* Introduction must be rewritten. Confuse, not precise, not clearly argumented.
The figure 1 has lot of interest but it has not its place in the introduction of this paper.
It could be introduced at the end of the paper to summarize the cellular targets of polyphenols (genistein, apigenin), artemisinin and sulforaphane, which were used.
The aim of the study is not clearly defined; genistein, apigenin, artemisinin and sulforaphane must be better presented. Few sentences on each molecules are required with additionnal refs..
* Methods OK including statistical analysis
* Data are clear and well presented clear
* Discussion, often very far from the data; the different parts of the discussion must be clarified.
* Conclusion, must be improved; the interest of the findings are not clearly summarized.
Whereas the data are clear, the overall presentation and organization of the paper deserves the interest of the study. The paper therefore needs major modifications.
Author Response
Comment 1: Title must be reconsidered. [The title is] not adapted, the interest of the paper is based on the effects of polyphenols, mainly genistein, as an anti-infectious drug to prevent COVID-19 infection as it acts on several parameters involved It is written in the abstract: Inflammation is associated with the pathophysiology of type 2 diabetes and COVID-19. Phytochemicals may modulate inflammation, expression of SARS-CoV-2 viral entry receptors (ACE2, TMPRSS2), and glucose transport in the gut. This study assessed how genistein, apigenin, artemisinin and sulforaphane affect these processes.
Response 1: We are not really sure what the reviewer is asking for here. However, we think that the title clearly reflects the results that are presented in the manuscript and that the revised Introduction and Discussion sections have made it more focused and aligned to the title.
Comment 2: Abstract must be reconsidered.
Response 2: The Abstract has been revised (lines 19-26).
Comment 3: Introduction must be rewritten. Confuse, not precise, not clearly argumented. What is the interest to use polyphenols: genistein, apigenin, artemisinin and sulforaphane? Positive control is missing for the parameters studied
Response 3: The Introduction has been revised to ensure a clearer focus on the rationale for our study, emphasising the key research questions and hypotheses that directly relate to our findings. Positive controls were included in the initial screening of compounds (Supplementary Table 1).
Comment 4: The figure 1 has lot of interest but it has not its place in the introduction of this paper. It could be introduced at the end of the paper to summarize the cellular targets of polyphenols (genistein, apigenin), artemisinin and sulforaphane, which were used.
Response 4: Figure 1 has been moved to the Discussion section (Figure 7) and summarises the effect of genistein on the target genes.
Comment 5: The aim of the study is not clearly defined; genistein, apigenin, artemisinin and sulforaphane must be better presented. Few sentences on each molecules are required with additional refs..
Response 5: The Introduction has been revised to define the study’s aim and better integrate the role of genistein, apigenin, artemisinin and sulforaphane in the research context (lines 72-77).
Comment 6: Discussion, often very far from the data; the different parts of the discussion must be clarified.
Response 6: The Discussion section has been revised to focus on our key findings, with greater emphasis on the interaction between ACE2 and glucose transporters.
Comment 7: Conclusion, must be improved; the interest of the findings are not clearly summarized.
Response 7: We appreciate the reviewer’s suggestion to improve the Conclusion. However, we respectfully disagree with the comment that the interest in our findings is not clearly summarised. We believe that the original conclusion effectively highlights the key outcomes and their implications.
Round 2
Reviewer 1 Report
None
None
Reviewer 3 Report
The paper is clear and wwell illustrated.
No modification are required.
As indicated in the last sentence of the abstract: "Genistein downregulated the inflammation-induced increase in SGLT1 and TMPRSS2,36 which may help lower postprandial glycaemic response and COVID-19 risk/severity in37
healthy individuals and those with metabolic disorders".
This is the main finding, and the data supporting this finding are clear and well illustrated.
These data bring new information on the activity of genistein in the context of diabetes and COVId19 susceptibility.
As indicated in the last sentence of the abstract: "Genistein downregulated the inflammation-induced increase in SGLT1 and TMPRSS2,36 which may help lower postprandial glycaemic response and COVID-19 risk/severity in37
healthy individuals and those with metabolic disorders".
This is the main finding, and the data supporting this finding are clear and well illustrated.
These data bring new information on the activity of genistein in the context of diabetes and COVId19 susceptibility.